# Evaluation of Anticancer Activity of Zhubech, a New 5-FU Analog Liposomal Formulation, against Pancreatic Cancer

**DOI:** 10.3390/ijms24054288

**Published:** 2023-02-21

**Authors:** Nkafu Bechem Ndemazie, Raviteja Bulusu, Xue You Zhu, Esther Kesewaah Frimpong, Andriana Inkoom, Joy Okoro, Dexter Ebesoh, Sherise Rogers, Bo Han, Edward Agyare

**Affiliations:** 1College of Pharmacy and Pharmaceutical Sciences, Florida A&M University, Tallahassee, FL 32307, USA; 2Faculty of Health Sciences, University of Buea, Buea P.O. Box 63, Cameroon; 3Department of Medicine, University of Florida, Gainesville, FL 32608, USA; 4Department of Surgery, Keck School of Medicine University of South California, Los Angeles, CA 90033, USA

**Keywords:** 5-FU, MFU, liposomal nanoparticles, Zhubech, Gd-Hexanoate, distribution

## Abstract

Pancreatic cancer is projected to be the second leading cause of cancer-related death by 2030 in the US. The benefits of the most common systemic therapy for various pancreatic cancers have been masked by high drug toxicities, adverse reactions, and resistance. The use of nanocarriers such as liposomes to overcome these unwanted effects has become very popular. This study aims to formulate 1,3-bistertrahydrofuran-2yl-5FU (MFU)-loaded liposomal nanoparticles (Zhubech) and to evaluate itsstability, release kinetics, in vitro and in vivo anticancer activities, and biodistribution in different tissues. Particle size and zeta potential were determined using a particle size analyzer, while cellular uptake of rhodamine-entrapped liposomal nanoparticles (Rho-LnPs) was determined by confocal microscopy. Gadolinium hexanoate (Gd-Hex) was synthesized and entrapped into the liposomal nanoparticle (LnP) (Gd-Hex-LnP), as a model contrast agent, to evaluate gadolinium biodistribution and accumulation by LnPs in vivo using inductively coupled plasma mass spectrometry (ICP-MS). The mean hydrodynamic diameters of blank LnPs and Zhubech were 90.0 ± 0.65 nm and 124.9 ± 3.2 nm, respectively. The hydrodynamic diameter of Zhubech was found to be highly stable at 4 °C and 25 °C for 30 days in solution. In vitro drug release of MFU from Zhubech formulation exhibited the Higuchi model (R^2^ value = 0.95). Both Miapaca-2 and Panc-1 treated with Zhubech showed reduced viability, two- or four-fold lower than that of MFU-treated cells in 3D spheroid (IC_50Zhubech_ = 3.4 ± 1.0 μM vs. IC_50MFU_ = 6.8 ± 1.1 μM) and organoid (IC_50Zhubech_ = 9.8 ± 1.4 μM vs. IC_50MFU_ = 42.3 ± 1.0 μM) culture models. Confocal imaging confirmed a high uptake of rhodamine-entrapped LnP by Panc-1 cells in a time-dependent manner. Tumor-efficacy studies in a PDX bearing mouse model revealed a more than 9-fold decrease in mean tumor volumes in Zhubech-treated (108 ± 13.5 mm^3^) compared to 5-FU-treated (1107 ± 116.2 mm^3^) animals, respectively. This study demonstrates that Zhubech may be a potential candidate for delivering drugs for pancreatic cancer treatment.

## 1. Introduction

Pancreatic cancer is projected to be the second leading cause of cancer death by 2030, after lung cancer, in the United States [1]. The management of pancreatic cancer is multifaceted, comprising surgery, radiotherapy, and chemotherapy, with the choice of treatment mainly dependent on the type and stage of the disease [2]. The benefits of chemotherapy, the most common systemic therapy for various cancers, have been masked by high drug toxicities, adverse reactions, and drug resistance [3,4]. Cancer burden worldwide has necessitated modern approaches that ensure safer, targeted, and efficient delivery of drugs to the tumor site. One of such approach is the use of nanoparticles for targeted drug delivery to overcome the challenge of tumors receiving low drug concentrations [5,6].

Nanoparticles range in size from between 1 and 1000 nm [7]. Their composition includes phospholipids and other polymeric materials [8]. The size and design of nanoparticles increase the ease at which they permeate tumors, enhancing the deposition of a higher concentration of therapeutic agents in the tumors. Nanoparticles work by targeting cancer cells, tumor environment, or the immune system [7]. Liposomes, dendrimers, and polymeric micelles are nanoparticles used to treat cancer and other disease conditions by acting as carriers for various drugs [4]. These nanoparticles offer the advantage of increased water solubility of drugs, targeted drug delivery, improved stability, better circulation time, and benefits in tumor imaging, ultimately making them suitable carriers in cancer therapy [3,4]. Nanoparticles have been around for over five decades. However, their use in cancer treatment has become more popular over the last three decades.

Liposomes, the first nanoparticle to be approved as a carrier in cancer therapy, were discovered about six decades ago [4,9]. Liposomes are spherical vesicles that comprise lipid concentric bilayers enclosing an aqueous core [10]. The lipid bilayer of liposomes resembles the bilayer of mammal cells, facilitating better interaction and cellular uptake [3]. Liposomes are classified based on their size and number of bilayers as either small or large unilamellar or multilamellar vesicles [7]. Some characteristics of liposomes that make them ideal candidates as nanocarriers are as follows: they are biodegradable and biocompatible; and they allow for the incorporation of both aqueous and lipid-soluble drugs, sustained drug action, and targeted drug delivery [11]. Doxil (liposomal-based Doxorubicin) was the first liposomal anticancer formulation approved by the Food and Drugs Authority (FDA) for treating AIDS-related Kaposi sarcoma and ovarian cancer [12]. The introduction of this novel formulation generated a heightened interest in liposome research, which subsequently led to the development and approval of other liposomal formulations. Such formulations include Daunoxome–Daunorubicin (indicated for AIDS-related Kaposi sarcoma and acute myeloid leukemia), Myocet-Doxorubicin (indicated for metastatic breast cancer), and Onivyde -Irinotecan (indicated for metastatic pancreatic cancer) [3].

Some liposomal formulations are currently undergoing clinical evaluation for cancer treatment. An example is EndoTAG-1 (liposomal Paclitaxel) for treating pancreatic cancer (the third deadliest cancer in the United States) [9,13]. Liposomes have been used in several studies as nanocarriers to determine the efficacy of novel and standard cancer drugs in various tumors [10]. Matsumoto et al. demonstrated the superior cytotoxic effects of a novel liposomal Gemcitabine formulation (FF-10832) on mouse xenograft tumor models over the unmodified drug in pancreatic cancer cells [14]. In that study, liposomal Gemcitabine significantly reduced tumor size in mice with Capan-1 and BxPC-3 pancreatic cancer tumors compared to Gemcitabine hydrochloride [14]. Additionally, Xu et al. demonstrated the potential of pH-sensitive liposomes in overcoming Gemcitabine resistance in pancreatic cancer management [15]. In addition, Inkoom and colleagues also demonstrated the effectiveness of Gemcitabine stearate nanoparticles in suppressing tumors using patient-derived xenograft mouse models [16]. Although they are novel methods for the preparation of liposomes, such as the chloroform injection and spontaneous phase-transition, film hydration still remains one of the most effective methods for preparation and high drug loading [17,18].

An approach to further increase the cytotoxicity of liposomes in tumors is by conjugating to high-molecular-weight compounds such as polyethylene glycol (PEG) [10,19]. The concept behind this approach is the enhanced permeability and retention effect (EPR), in which high-molecular-weight compounds accumulate in tissues with high vascular permeability, such as cancer tissues [10]. This accumulation results in increased bioavailability, sustained drug action, and higher cytotoxic effects. Kim et al. conducted a study comparing the efficacy of free cromolyn to pegylated liposomal cromolyn (PEG-lipo-cro) in BXPC-3 tumor-bearing mice [20]. PEG-lipo-cro significantly inhibited tumor growth in comparison to free cromolyn (*p* < 0.01) due to an increase in the compound’s half-life, leading to higher cytotoxicity [20]. In addition, PEGylation offers systemic stealth effects due to surface hydration to facilitate enhanced delivery to the target site [21].

Many novel cancer drugs never make it to clinical use due to differences in efficacy and safety data between animal and human studies. Traditional cancer models or two-dimensional (2D) models have a major shortcoming in their inability to mimic the actual tumors effectively. This has led to inconsistencies in in vitro, in vivo, and human study data, subsequently accounting for the deficient number of cancer drugs that make it to clinical use. Organoids were introduced into cancer research to better mimic patients’ tumors and to increase the likelihood of more drugs moving from clinical trials to bedside treatments [22]. Cancer organoids are three-dimensional (3D) models of cancer cells that mimic the morphological and histopathological features of patient tumors [23,24] Organoids have improved understandings of the heterogeneous nature of tumors and have also somewhat overcome the shortcomings of two-dimensional (2D) cancer models [23]. For example, pancreatic cancer organoids have been used to demonstrate different sensitivities to various chemotherapeutic agents (76 drugs), initially undocumented in 2D models, further substantiating the benefits of organoids in personalized therapy [25].

Shortcomings of standard chemotherapeutic agents have led to increasing research involving modified forms of these drugs. One common approach is conjugating long-chain hydrocarbons to traditional medicines, leading to the formation of lipophilic prodrugs [26,27]. Conjugation confers properties such as increased membrane permeability, bioavailability, and extended duration of action, ultimately leading to higher cytotoxic effects [16]. The present study details the following: the design of liposomal formulation (LnP); cell-viability studies using a previously synthesized 5-FU analog (1,3 bistetrahydrifuran-2yl-5-FU (MFU)) [28] against Panc-1 (with 2D and 3D spheroids) and MiaPaCa-2 cancer cells using 2D and 3D models (spheroids and organoids); and tumor-efficacy studies using a patient-derived xenograft (PDX) mouse model with ectopic tumors and evaluating the tissue biodistribution of Gd-Hex-LnPs in vivo.

## 2. Results

### 2.1. Formulation and Characterization of Zhubech

#### 2.1.1. Characterization of Zhubech

The LnP was made with a modification of a previously documented liposome [29,30] using DPPC, MPPC, cholesterol, and DSPE-PEG_2000_ at a molar ratio of 50:25:20:5. The mean particle size of the blank liposomal nanoparticle, LnP, was 90.0 ± 0.65 nm, (mean ± SD), while the mean size of Zhubech was 124.9 ± 3.2 nm (Appendix A). With our formulation technique, the entrapment efficiency (EE) of MFU was 97.2 ± 2.1% and a final zeta potential value was −30.3 ± 12 (Appendix A), as shown in Table 1.

#### 2.1.2. In Vitro Formulation Stability

There are many parameters used to assess in vitro stability, such as particle size (or hydrodynamic diameter), poly dispersity index, physical appearance, entrapment efficiency (or drug content), and zeta potentials. However, in this study, we evaluated the hydrodynamic diameter, physical appearance, and drug content (MFU) of zhubech at the following temperatures; 4 ± 2 °C, 25 ± 2 °C, and 40 ± 3 °C for 30, 60, and 90 days, respectively [31,32,33,34]. All formulations were stored in 20 mL glass vial containers during the study period. After 30 days (batch 1), all the liposomal formulations stored at all temperatures had normal physical appearances (cream white, clear). After 60 days (batch 2), all the formulations stored at 4 ± 2 °C and 25 ± 2 °C appeared cloudy and with suspended particles, while those stored at 40 °C appeared cream white but clear. However, 90 days later (batch 3), formulations stored at 4 ± 2 °C appeared cloudy, while those stored at 25 °C appeared suspended; however, the formulations stored at 40 °C appeared clear (Figure 1 and Table 2).

The particle size distribution (hydrodynamic diameter) of all the formulations as a function of temperature were also evaluated after 30, 60, and 90 days in three batches as shown in Table 2 (Appendix A). An increase in particle size was noted at a higher temperature (40 °C) in batches 2 and 3, while a decrease in drug content of more than 5% was observed in batches 2 and 3. These observations suggest the likelihood of leakage of content from the formulation after a 60- and 90-day period. In addition, at temperatures of 4 °C and 25 °C, we noticed a small decrease in drug content in batches 2 and 3, suggesting slight leakage (<1.2%) of the formulations at those temperatures (Table 2). Overall, the size distribution of the Zhubech formulation stored at all temperatures showed an insignificant decrease (<5%) in MFU content at 4 ± 1 °C and 25 ± 2 °C; however, a significant loss in MFU content (>10%) was observed in batches 2 and 3 at 40 ± 3 °C.

#### 2.1.3. In Vitro Drug Release Kinetics from Zhubech Formulation

Figure 2 shows the cumulative in vitro drug release of MFU from Zhubech over a 24 h period at 37 °C, while maintaining sink condition. The release of free MFU was rapid and almost complete from the dialysis bag, with about 80% released within just 2 h and about 90% released within the first 4 h. Free MFU release continued until it reached 95% within the first 6 h. In comparison to MFU release from Zhubech, about 50% of MFU was released within the first 2 h, followed by 70% release within 6 h, reaching steady state by 8 h. This suggests that any free MFU placed in the dialysis bag would rapidly diffuse significantly within a short period. In addition, its release profile could be used to differentiate the release pattern of MFU from Zhubech, as long as the sink condition was maintained and the diffusion of MFU to the receiving medium was dependent on the concentration gradient (Figure 2a). This implies that most of the MFU remained entrapped in Zhubech under the study conditions (Figure 2a).

To determine the MFU diffusion mechanism, in vitro release kinetics of MFU from Zhubech was modeled as shown in Figure 2b for zero-order kinetics, Figure 2c for first-order kinetics, and Figure 2d for the Higuchi model. The best-fit release mechanism was determined based on the R^2^ values of the various kinetic models (Figure 2b–d) [35]. The R^2^ value (0.97) of the Higuchi model (Figure 2d) was found to be the highest compared with the R^2^ value (0.84) of zero-order kinetics (Figure 2b) and the R^2^ value (0.92) of first-order kinetics (Figure 2c). This suggests that MFU released from LnP followed Higuchi diffusion kinetics and implies that MFU release comes from a homogeneous delivery system (conjugation) and diffuses out of the delivery system over a period of time [35,36].

#### 2.1.4. In Vitro Cellular Uptake

Lipid-based nanoparticles can be efficiently internalized by cancer cells and are hence an attractive system to deliver drugs to cancer cells. Cancer cells are known to efficiently uptake nanoparticles among different cell types and may act as a reservoir for nanotherapeutics for drug delivery [37,38]. To determine the ability of LnPs to enter the cells to deliver MFU, Panc-1 cells were incubated with rhodamine (Rho-14)-entrapped liposomes for 12 h and 24 h. Figure 3a shows that Panc-1 cells internalized LnP in a time-dependent manner. A confocal microscope was used to visualize the uptake of Rh0-14-labeled liposomes. The images and a video taken further demonstrated the orientation of the dye-loaded nanoparticles in a 3D style, showing the location of the rhodamine-liposomal formulation in the cytoplasm (Figure 3b and Appendix A). The accumulation of the labeled liposomes in perinuclear vesicles suggests the accumulation of formulation in the endosomal/lysosomal compartment in a time-dependent manner.

### 2.2. Cytotoxic Effect of Zhubech on MiaPaCa-2 and Panc-1 Cell Lines

Prior to conducting cytotoxicity to compare free MFU and Zhubech, studies were conducted to compare the efficacy of free MFU compared to 5-FU. This prior study showed superiority of MFU to 5-FU in 2D and 3D spheroids using MiaPaca-2 and Panc-1 cell lines (Figure 4).

The cytotoxic activity of Zhubech was compared to free MFU in MiaPaca-2 cells (2D and 3D spheroids and organoids) and Panc-1 cells via 2D and 3D (spheroid) culture models, using alamar blue assay [39]. As shown in Figure 5, Zhubech demonstrated significant cytotoxic activity against the 2D Panc-1 culture with an IC_50_ value of 2.0 ± 1.1 μM compared to free MFU (IC_50_ value of 3.4 ± 1.1 μM) after 48 h treatment (Figure 5a,b). In addition, Zhubech-treated and free-MFU-treated 3D Panc-1 cultures recorded IC_50_ values of 3.4 ± 1.0 μM and 6.8 ± 1.1 μM, respectively (Figure 5a,b). These findings are similar to results obtained from our previous studies [28]. We noted that IC_50_ values were expectedly higher in the treated organoids of the MiaPaCa-2 culture. Zhubech-treated organoids of the MiaPaCa-2 culture showed IC_50_ values (IC_50_ = 9.8 ± 1.4 μM) that were four-fold lower than the free-MFU-treated organoid MiaPaCa-2 culture (IC_50_ = 42.3 ± 1.0 μM) (Figure 5c). Table 3 below compares the IC_50_ values of Panc-1 and MiaPaCa-2 cells using 2D and 3D spheroids and organoids.

### 2.3. Tumor-Efficacy Studies

Mice bearing pancreatic PDX tumors were treated over a period of thirty-six days. In the Zhubech-treated group, significant tumor growth suppression was observed compared to the untreated control and 5-FU-treated groups (Figure 6a). Tumor suppression was noticed up to two weeks, after which the 5-FU group noticed a geometric increase, while the Zhubech-treated group noticed a decrease in tumor volume for the rest of the study period. The mean tumor volume of the untreated control group was extremely large, while the mean tumor volume of Zhubech-treated group (108 ± 13.5 mm^3^) exhibited significantly lower tumor growth compared with the mean tumor volume of the 5-FU-treated group (1107 ± 116.2 mm^3^), especially by week 5. Figure 6b shows the change in the weight of the mice over the study period, which is a representation of the toxicity of the drugs/compound in the animal. There is an insignificant change in the weight of the mice between during the study period, which is less than 10%, suggested to be due to the toxicity of a drug [40]. All of these are similar to findings obtained from our previous studies which showed superiority of MFU to GemHCl [28].

### 2.4. Tissue Biodistribution of Gd-Hex-LnP

Figure 7 shows the amount of Gd-Hex-LnP deposited in various tissues at different time points. We used Gd-Hex-LnP in place of MFU to enable easy quantification using inductively coupled plasma mass spectrometry (ICP-MS). There was overall a significant amount of Gd-Hex-LnP deposited per gram of wet tissue in the liver, kidney, lungs, and pancreas. In the heart and brain, the amount of Gd-Hex-LnP deposited at each time point was insignificant compared to the other tissues, such as the lungs, kidneys, pancreas, and liver. In the kidney and lungs, there was a significant deposition noted at 4 h and 1 h, respectively. The liver and the pancreas each noted a significant increase at two time points, with the liver, increase occurred at 0.5 h and 1 h, while the pancreas had a significant increase at 1 h and 4 h (Figure 7).

## 3. Discussion

Pharmaceutical nanoparticle delivery systems are ideal for transporting anticancer drugs and reducing unwanted distribution and side effects to healthy cells. These nanoparticles protect anticancer drugs from first-pass metabolism and enzymatic degradation. In addition, these delivery systems generally improve the drug’s enhancement properties, prolong systemic circulation, and increase drug entrapment and loading capacity [3,41]. Furthermore, nanoparticles as anticancer drug delivery systems are generally designed to improve, for example, high drug loading capacity, prolonged systemic circulation, the ability of the nanoparticle to accumulate specifically in the required pathological zone, and the nanoparticle’s ability to resist degradation in high environmental temperatures [42].

To resolve some of the issues mentioned above, we synthesized and characterized MFU from our previous studies [28], loaded the LnP with it (together called Zhubech) using a surface modified with /DSPE-PEG_2000 to enhance stability in systemic circulation, avoiding the reticuloendothelial system through the stealth effect, and improving the therapeutic efficacy of MFU. One of the unique features of Zhubech is that it is very stable at room temperature during the first 30 days; hence, it might not need special temperatures for storage during its use [43]. Although they are novel methods of preparation of liposomes, such as the chloroform injection and spontaneous-phase-transition methods, we used film hydration to prepare our liposomes as we were interested in preparing the formulation rather than optimizing production [17,18]. The LnP was prepared using the film hydration method with DPPC, MPPC, cholesterol, and DSPE-PEG_2000. Furthermore, particle size <200 nm has been reported to have a considerable decrease in the leakage of entrapped lipophilic content, such as MFU, due to conjugation in the lipids [44]. The release characteristics of MFU from Zhubech follows the Higuchi release model, suggesting that the formulation Zhubech behaves like a matrix system with release characterized by diffusion [45]. This assumes that the initial drug concentration in the matrix is much higher than drug solubility, with drug diffusion taking place only in one dimension, drug particle much smaller than system thickness, and matrix swelling and dissolution being negligible. Further assumptions are that drug diffusivity is constant with perfect sink condition in place [46].

In an in vitro cell-viability study, we assessed the effects of MFU and 5-FU on MiaPaca-2 and Panc-1 cell lines in 2D and 3D spheroid models, which demonstrated the superiority of MFU compared to the parent drug 5-FU in cytotoxicity. We also demonstrated the effects of MFU and Zhubech on Panc-1 (2D and 3D spheroid models) and MiaPaca-2 (3D organoid model) cell lines. Cells were exposed to the blank liposome (LnP), MFU, or Zhubech for 48 hr, a sufficient period to assess inhibition. The LnP was well tolerated by Panc-1 cells in 2D and 3D spheroid cell cultures, as well as MiaPaca-2 3D organoids. MFU and Zhubech showed a dose-dependent inhibitory effect with Zhubech demonstrating a more significant reduction in cell viability in all culture models. This suggests a higher cellular uptake of liposomal form, leading to a greater internalization of MFU.

In vivo anti-tumor activity of Zhubech after the 15th day exhibited a significant inhibition of tumor growth compared with 5-FU and the untreated control. There was a significant decrease in tumor volume for Zhubech-treated mice (108 ± 13.5 mm^3^) compared to 5FU-treated mice (1107 ± 116.2 mm^3^) at the end of the studies (*p*-value < 0.001). This extraordinary tumor efficacy of Zhubech could be explained by: the inability of the dihydropyrimidinone dehydrogenase (DPD) enzyme to metabolize MFU in LnP due to the absence of a free -NH2 group, as well steric hindrance from the THF-functional group on free MFU. This likely allowed MFU to circulate longer, be more bioavailable, and have better therapeutic results. Additionally, the conjugation of THF to 5-FU may have given MFU some degree of lipophilicity, which may have made it easier for it to reach cancer cells. Lastly, the loading MFU in Zhubech might have increased its half-life, increasing permeability to the tumor site due to the stealth effect and improving stability and duration in circulation.

Nanocarriers are very useful in drug delivery because of their ability to alter pharmacokinetics and biodistribution. For the biodistribution study, Gd-Hex delivered to a mouse by an LnP (Gd-Hex-LnP) was significantly higher in the liver, lungs, and pancreas after 1 h and 4 h, respectively. The mechanism of delivery by LnP could be attributed to higher enhanced permeability and retention effect where nanoparticle size carriers are distributed in tissues with a rich blood supply and are retained there at high concentrations for a long period. In contrast, free drugs are not retained and instead return to circulation by diffusion [47,48,49]. In addition, environmental temperatures did not influence the stability of Zhubech (Table 2), possibly improving the delivery of Gd-Hex to tissues [50,51].

## 4. Materials and Methods

### 4.1. Materials

1,2-distearoyl-sn-glycero-3-phosphoethanolamine-N-[amino(polyethyleneglycol)-2000] (DSPE-PEG_2000_), Dipalmitoylphosphatidylcholine (DPPC), 1-Myristoyl-2-palmitoyl-sn-glycero-3-phosphocholine (MPPC), cholesterol (chol) lipids were all purchased from Avanti Polar Lipids, Inc. (Alabaster, AL, USA). All the chemicals, including 5-FU and reagents, were purchased from Sigma-Aldrich (St. Louis, MI, USA). Pca cell lines (Panc-1 and MiaPaca-2) were obtained from the American Type Culture Collection (ATCC). All other chemicals used were of an analytical reagent grade.

### 4.2. Preparation of Stealth LnP

The LnP was prepared using the thin-film hydration technique [52]. Briefly, MPPC, DPPC, Chol, and DSPE_PEG-2000 were dissolved in 2 mL of chloroform and mixed at a molar ratio of 25:50:20:5 in a round-bottom flask. The chloroform was removed using a rotary evaporator at 75 °C for 30 min until a thin, dry film was formed. Samples were then placed under a high vacuum where the lipid sample formed a “swollen” film that was held under vacuum for 2–3 h to remove residual chloroform. These dried, swollen lipid films were hydrated with phosphate-buffered saline (PBS), in which the drug was dissolved and held at 65–70 °C (slightly above the transition temperature) and homogenized using a NanoDeBEE homogenizer at a pressure of 30,000 PSI for 20 cycles, before being sonicated for 5–10 min. Then, the formulation was extruded 10 times through stacked polycarbonate filters with a pore size of 100 nm (Nuclepore Track-Etch membrane) at 65–70 °C using an extruder (Avanti^®^, Birmingham, AL, USA).

### 4.3. Physical Characterization

#### 4.3.1. Particle Size, PDI, and Zeta Potential Determination

The particle size (hydrodynamic diameter) of the liposomes was determined by dynamic light scattering using a particle sizer (NICOMP ™ 380 ZLS, Santa Barbara, CA, USA). All measurements were performed at room temperature. Before each measurement, the samples were diluted with deionized water at a ratio of 1:10. The polydispersity index (PDI) and Zeta potential were measured using the same instrument (NICOMP 380 particle sizer) [53]. Zhubech formulations stored in 20 mL borosilicate vial glasses were divided into 3 batches (corresponding to the duration of study) and kept at different temperatures for modified stability studies at 1, 2, and 3 months to measure hydrodynamic diameter, MFU content in Zhubech, and physical appearance [27,54,55].

#### 4.3.2. HPLC Analysis

The Waters HPLC alliance e2695 system with a PDA detector was used to analyze MFU at a wavelength of 270 nm. The mobile phase was water (pH adjusted to 2.5 with phosphoric acid) and methanol at a ratio of 90:10 (*v*/*v*). The mobile-phase flow rate was maintained at 1.0 mL/min. MFU retention time was 19.6 min. The injection volume was 50 μL. Data acquisition and analysis were performed using empower software (Waters Corporation, MA, USA). The calibration curve (peak area vs. concentration) was generated over the range of 1–100 μg/mL and was found to be linear with a correlation coefficient of 0.9998. Before analysis, the reverse phase column was equilibrated with the mobile phase made up of water and methanol in a ratio of 90:10, and the pH was adjusted to 2.5. Isocratic elution was performed throughout the entire analysis, including internal standards.

#### 4.3.3. Entrapment Efficiency

The supernatant obtained after centrifugation of Zhubech was analyzed for unentrapped MFU by high-performance liquid chromatography (HPLC, Waters, USA) using a C18 column. The mobile phase consisted of H_2_O at a pH of 2.5 and methanol at *v*/*v* of 90:10. The injection volume was 50 μL, and the flow rate was 1 mL/min. The *EE*% was calculated according to the following equation:$$EE\%=\frac{Total\;drug-Free\;drug}{Total\;drug}*100$$

#### 4.3.4. Release Studies

The release of MFU from Zhubech was studied using a dialysis method. The dialysis bags were soaked in distilled water at room temperature for 12 h to activate the dialysis bag and to remove the preservative, and then were rinsed thoroughly in distilled water.

In vitro release of the MFU from Zhubech was performed by dialysis in a dialysis bag (14,000 MW cut off; Sigma-Aldrich) containing 150 mL of phosphate-buffered saline and methanol at a ratio of 3:1 *v*/*v* (PBS; pH 5.6). Two bags were prepared containing Zhubech and the control containing free MFU. Equivalent amounts of Zhubech and free MFU concentrations were added to a dialysis bag. Both bags were prepared and tested together with liposomal dispersions. The bags were suspended in a suitable solvent flask so that the part of the dialysis bag containing the formulation was immersed in a buffer solution (PBSl). The flask was held on a magnetic stirrer (Matrex), and stirring was maintained at 200 rpm and a temperature of 37 °C. Samples were collected at 0.5, 1, 2, 4, 6, 8, 12, and 24 h, while maintaining sink conditions throughout [56].

#### 4.3.5. Release Models

The release kinetics of 5-FU from the heat-sensitive liposomes were investigated to predict the possible mechanism of release using mathematical models. The release order was determined using zero-order Equation (1a) and the first-order kinetic model, as shown below.
C = C_0_ + K_0_t (1a)
LogC = LogC_0_ + K_1_t/2.303(1b)
where C_0_ is the initial amount of drug, C is the % cumulative MFU released (zero order) or first-order Equation (1b) at time “t”, and K_0_ is the zero-order release constant, and K_1_ is the first-order release constant.

The Korsmeyer–Peppas Equation (2) and Higuchi Equation (3) models were used to determine whether the release mechanism follows the polymeric system (power law) or Fickian diffusion, as shown below:C_t_/C_∞_ = Kt^n^(2)
C = C_0_ + K_H_t^1/2^(3)
where C_t_/C_∞_ is the fraction of drug release at time, t, K is the rate constant, and n is the release exponent. Meanwhile for the Higuchi model, C is the % cumulative MFU release at the time, and t and K_H_ is the Higuchi constant.

### 4.4. Cell-Viability Studies

These in vitro viability studies were performed on pancreatic and colorectal cancer cell lines. MiaPaca-2 was incubated in DMEM with high glucose, and L-glutamine and Panc-1 were incubated in McCoy 5A modified media, both supplemented with 10% fetal bovine serum (FBS), 1% penicillin-streptomycin (PenStrep), and 2.5% 4-(2-hydroxyethyl)-1-piperazineethanesulfonic acid (HEPES). Cells were plated in a T75 flask for culture and later seeded into 96-well plates.

#### 4.4.1. 2D-Cell-Viability Studies

MiaPaca-2 and Panc-1 cells were seeded in 96-well plates at a density of 5000 cells/well and incubated in 5% CO_2_ and at a temperature of 37 °C. A stock solution of Zhubech and free LnP were prepared in molar concentrations and serially diluted with the growth medium to prepare different concentrations, i.e., 80, 40, 20, 10, and 5μM. All cells were treated with 200 μL of each drug concentration in quintuplicate and incubated for 48 h. At the end of the treatment, 20 μL of 0.15% resazurin sodium salt (Alamar blue^®^) was added and incubated for four hours under optimal conditions (5% CO_2_, 37 °C). Fluorometric analysis was determined at an excitation wavelength of 560/580 nm and an emission wavelength of 590/610 nm, and the percentage of viable cells per concentration was calculated.

#### 4.4.2. 3D-Cell-Viability Studies

MiaPaca-2 and Panc-1 cells were seeded in a special Nunclon Sphera^®^ 96-well plate at a density of 5000 cells/L, and then 100 μL of fresh complete media was added to obtain 200 μL in each well. Then, the plates were centrifuged at 1500 rpm and incubated at 5% CO_2_ and a temperature of 37 °C for 24 h to yield spheroid formation. During treatment, 100 μL of the supernatant was replaced with the drug in the growth medium prepared as described in 2D viability studies. At the end of the treatment, 50 μL of 0.15% resazurin sodium salt (Alamar blue^®^) was added to each well and carefully dispersed by pipetting, before being incubated for 4 h. The drug was then added to the growth medium. Fluorometric analysis was measured as described above.

#### 4.4.3. Organoid Cell Viability

The MiaPaca-2 cell pellets were suspended in collagen and Matrigel (transglutaminase) mixture (20:1), as previously documented [57]. Then, the cells were transferred to a 48-well plate and incubated at 5% CO_2_ at 37 °C for 40 min. The Matrigel (transglutaminase) helps solidify collagen through cross-linking to form organoids. Then, 10 μL of the mixture was seeded into the 48-well plate and rested in the incubator for 40 min to solidify. Once the mixture was solidified, it was given to the supplements as media for 24 h. The organoids were allowed to develop over 48 h, followed by treatment with Zhubech and free MFU as the control over 7 days. The organoids were digested with 0.25% trypsin over 40 min, and an MTT assay was used to quantify cell viability.

### 4.5. Cellular Uptake Studies

Panc-1 cancer cells were grown in 6-well plates (with coverslips) at a cell density of 2 × 10^3^ for 24 h at 37 °C. The cells were then treated with Rhodamine-labeled LnP in growth media. After 12 and 24 h, Rhodmaine-LnP was removed, and the cells were gently washed twice with PBS. Next, 5 μg/mL of DAPI dye was added for nuclear staining; the cells were fixed using 4% paraformaldehyde, then mounted and imaged using the Leica SP2 Multiphoton system [29].

### 4.6. Animal Study

Ethics statements: Eight-week-old mice were obtained from the Jackson Laboratory (Bar Harbor, ME). The mice were housed in a virus-free environment that was temperature-controlled with indoor light, and they were provided access to food and water ad libitum for one week before treatment started. All procedures with mice were in strict accordance with the National Institutes of Health Guide for the care and Use of Laboratory Animals and the Animal Research Reporting of In Vivo Experiments (ARRIVE) guidelines. This was approved by the Florida A&M University Animal Care and Use Committee.

Tumor transplantation: Tumor tissue was surgically implanted in the left flank of immunocompromised mice as previously described [58]. A viable portion of resected tissue was isolated immediately following resection of primary PCa specimens to minimize critical ischemia time. The PCa tissue was then implanted subcutaneously into 8-week-old mice (*n* = 15). Xenografts were allowed to grow to a maximum of 1.5 cm before implantation to the flank of the new host.

Tumor-efficacy studies: In this study, mice bearing surgically implanted tumors were randomized into groups, namely the control, 5-FU, and Zhubech groups (*n* = 5/group), once tumor volumes became palpable and reached a range of 70–100 mm^3^. Baseline tumor volumes were established, and dosing initiation began with intravenous administration of 40 mg/k 5-FU and Zhubech (with 5-FU equivalent doses) twice weekly for six weeks. Tumor measurements were performed every other day. Tumor volumes were measured using calipers and were calculated using the following equation: V = (L*(W)^2^)/2. V is the volume (mm^3^), W(width) is the smaller of two perpendicular tumor axes and the value L (length), which is the larger of two perpendicular axes. Tumor growth volumes were calculated for each treatment group.

Euthanization: Carbon dioxide (CO_2_) flow to the chamber was adjusted to 3 L per minute for 2 to 3 min and each mouse was observed for lack of respiration and faded eye color. The CO_2_ flow was maintained for a minimum of 1 min after respiration ceased, followed by decapitation with scissors. The tumors were then incised and prepared for immunohistochemistry studies.

#### 4.6.1. Synthesis of Gadolinium Hexanoate (Gd-Hex) LnP

A mixture of GdCl_3_ (2.88 g, 10.9 mmol), Hexanoic acid (3.8 g, 32.7 mmol), and ammonium hydroxide (4 mL) in H_2_O (20 mL) was heated at 80 °C for 2 h. The white solid was collected and washed with acetone (20 mL), Et_2_O (20 mL). The solid was dried under a vacuum for 24 h. The same technique for Zhubech formulation was used to prepare Gd-Hex-LnP formulation and particle size was measured (Appendix A).

#### 4.6.2. Tissue Biodistribution

Nude mice (*n* = 18) were allowed to acclimatize for one week, which was followed by the studies. Mice were grouped into six groups (*n* = 3 each), corresponding to each time point (10, 30, 60, 240, 480, and 720 min). Mice received 0.05 mmol/kg Gd-Hex-LnP intravenously through the tail vein and were sacrificed at various time points. The tissues obtained and weighed.

After weighing, the organ samples were pre-treated with aqua regia (HNO3: HCl) in a 3:1 *v*/*v* ratio and allowed to stand overnight in a fume hood; the pre-treated samples were further diluted with 2% nitric acid (70%, *v*/*v*) in distilled water, centrifuged at 4000 rpm for 10 min, and finally filtered to remove debris [30]. The final sample solutions were analyzed by inductively coupled plasma mass spectrometry (ICP-MS) to determine the quantity of Gd in each sample solution.

### 4.7. Statistical Analysis

All results are presented in the form of means ± SEM. The difference between the MFU and Zhubech treatment groups was analyzed using ANOVA and, where necessary, significance was considered for *p* values < 0.05. All experiments were performed in at least triplicate, and analysis was conducted using GraphPad Prism 5.0 software (GraphPad Software, Inc., San Diego, CA, USA). When necessary, results were presented in simple tables, graphs, and bar charts.

## 5. Conclusions

We successfully formulated liposomal nanoparticles, LnPs, exhibiting enhanced release of MFU under physiologic conditions. The Zhubech nanoparticles exhibited a more potent anticancer effect compared to MFU against Panc-1 and MiaPaca-2 cancer cells. In addition, Zhubech showed superiority to 5-FU in vivo in terms of efficacy in tumor growth. These findings provide strong evidence in support of a possible therapeutic application of Zhubech as a drug delivery system for MFU, which can overcome some of the limitations of MFU, such as poor retention and short half-life. Future studies will look into completing stability and characterization through lyophilization, measuring zeta potentials, conducting flow cytometry to quantify colocalized dye in LnP, using transmission electronic microscopy (TEM) and scanning electronic microscopy (SEM) to measure particle size, and finally targeting Zhubech for in vitro and in vivo studies. Nevertheless, this study was not without limitations, such as the inability of Panc-1 cells to form organoids.

## Figures and Tables

**Figure 1 ijms-24-04288-f001:**
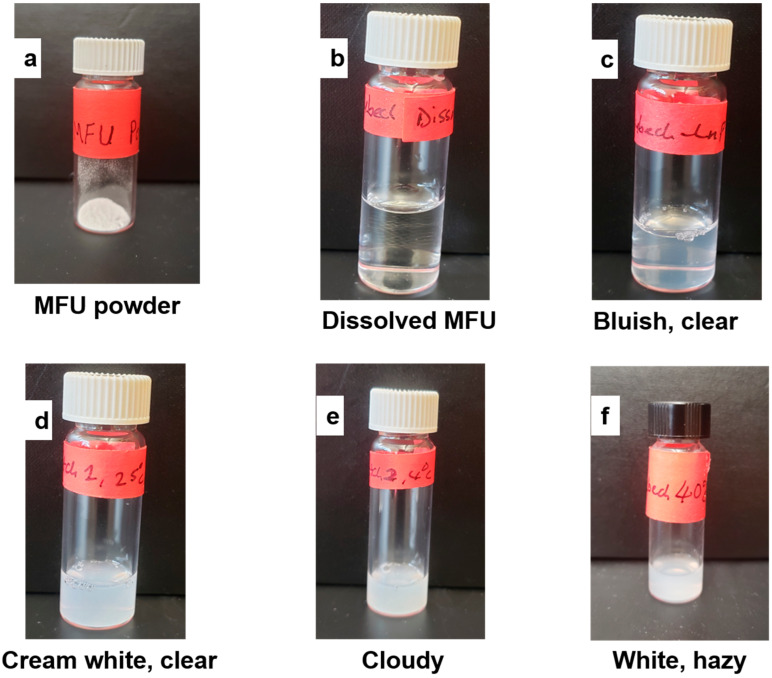
Physical appearance of Zhubech stored in (20 mL borosilicate clear Type 1, class A) glass vials over time. (**a**) Powdered appearance of MFU after synthesis, (**b**) MFU dissolved in distilled water to form a solution, (**c**) bluish, clear appearance after formulating Zhubech, (**d**) cream white, clear appearance of batch 1 formulation at all temperatures, (**e**) cloudy appearance of batches 2 and 3 stored at 4 °C, and (**f**) white hazy appearance of batches 2 and 3 stored at 40 °C.

**Figure 2 ijms-24-04288-f002:**
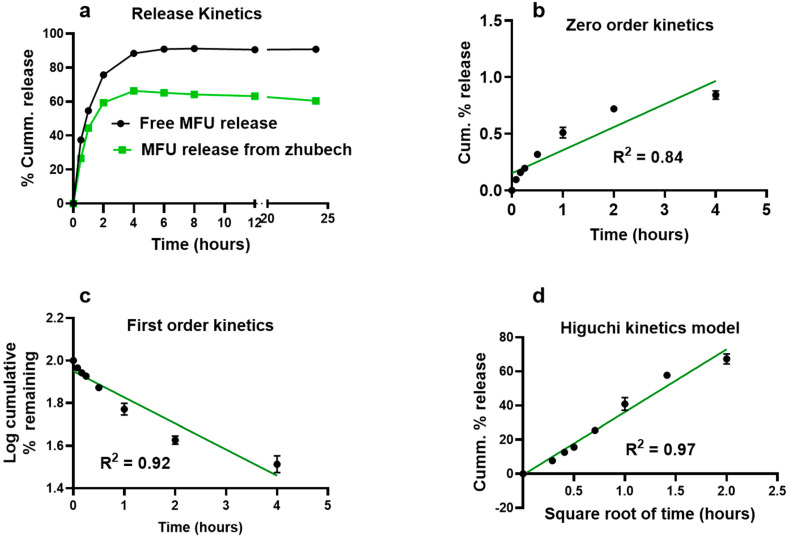
In vitro release kinetics of MFU from Zhubech. (**a**) In vitro cumulative release of free MFU and MFU from Zhubech at 24 h. (**b**) Release profile of MFU from Zhubech exhibiting zero-order kinetics; the release followed a non-linear pattern with R^2^ = 0.84. (**c**) Release profile of MFU showing first-order kinetics; the release followed an inverse linear pattern with R^2^ = 0.92. (**d**) Release profile of MFU exhibiting Higuchi release model; the release followed a linear pattern with R^2^ = 0.97.

**Figure 3 ijms-24-04288-f003:**
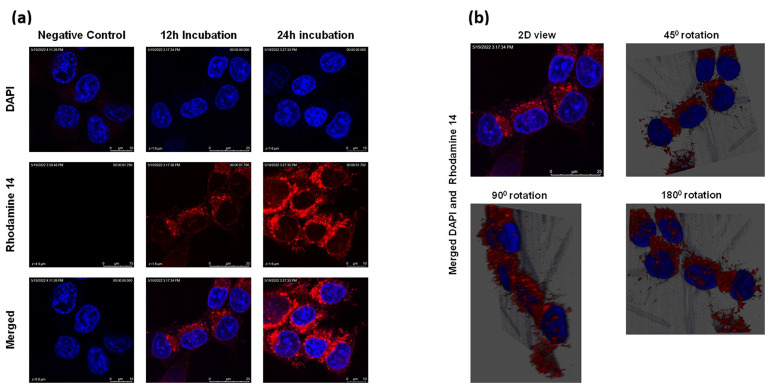
Cellular uptake of LnP. (**a**) 2D view of the cellular uptake of Rho-14-loaded LnP in Panc-1 cells over 12 h and 24 h with DAPI used as a nuclear stain. (**b**) 2D image of the 12 h incubation image of Rho-14, rotated to show the localization of LnP in 3D view. (See 25 s video of complete rotation in Appendix A).

**Figure 4 ijms-24-04288-f004:**
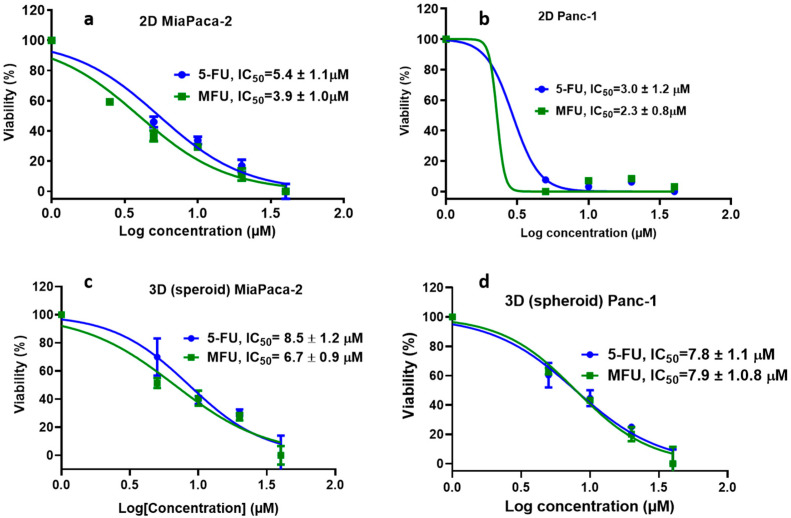
In vitro studies comparing the efficiency of 5-FU and free MFU. (**a**) 2D culture of MiaPaca-2 cells, (**b**) 2D culture of Panc-1 cells, (**c**) 3D culture of Miapaca-2 cells, and (**d**) 3D culture of Panc-1 cells.

**Figure 5 ijms-24-04288-f005:**
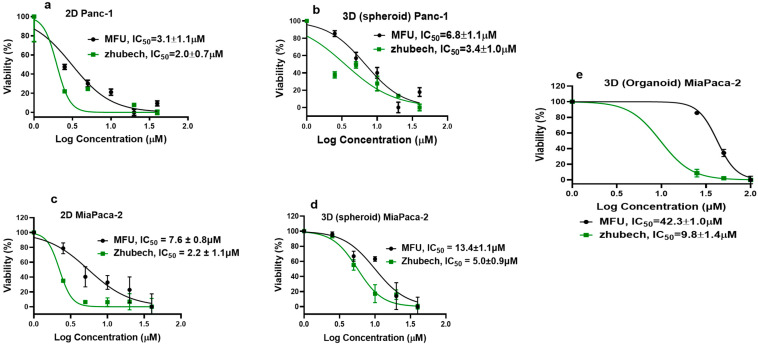
Cell viability: effects of Zhubech formulation on Panc-1 and MiaPaca-2 cell growth after 48 h exposure. (**a**,**b**) % viability of Panc-1 in 2D and 3D spheroids culture, respectively; (**b**–**d**) % viability of MiaPaca-2 in 2D and 3D spheroid culture respectively; (**e**) % viability of MiaPaca-2 in 3D organoid culture (after 7 days of exposure with MFU and Zhubech formulation).

**Figure 6 ijms-24-04288-f006:**
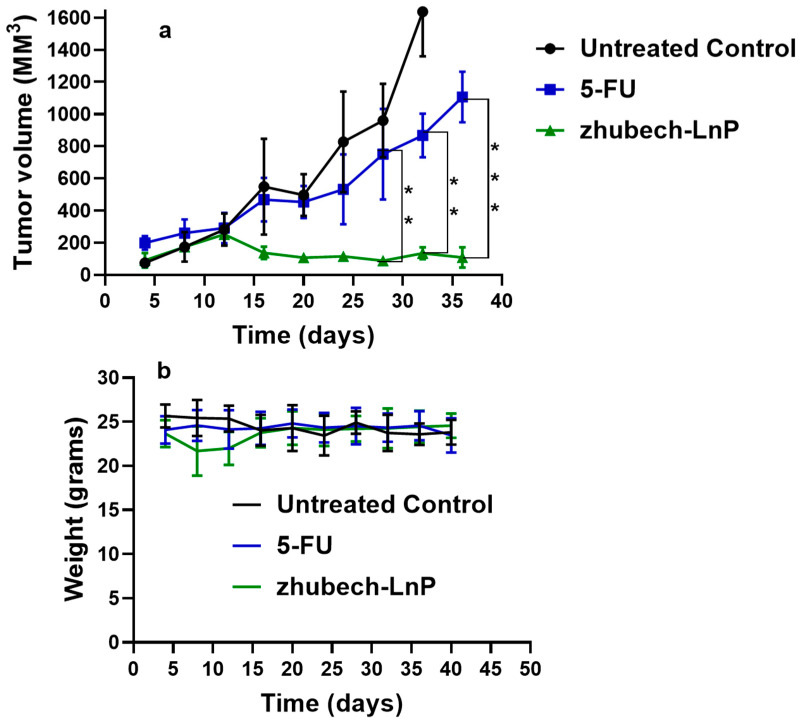
In vivo efficacy of Zhubech in PDX mouse model and chronic toxicity. (**a**) Tumor growth curves of 5-FU and Zhubech-treated mice bearing pancreatic PDX tumor and (**b**) body weight during treatment. Asterisks represents level of significance between control and treatment group (** *p* < 0.01, *** *p* < 0.001). All data represents mean ± SD, (*n* = 4/group).

**Figure 7 ijms-24-04288-f007:**
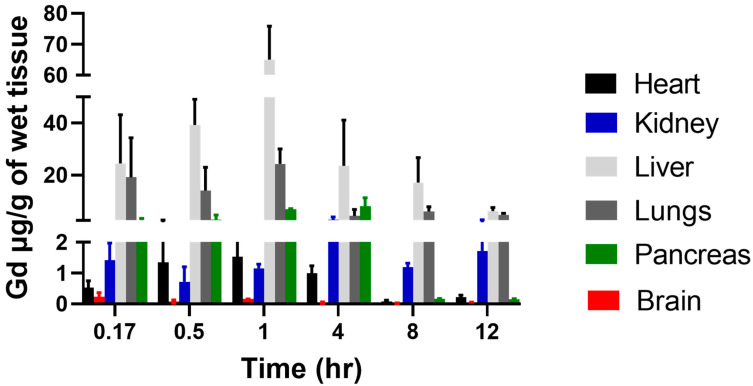
Biodistribution of Gd-Hex-LnP in different tissues (heart, kidney, lungs, brain, liver, and pancreas) at different time points (0.17, 0.5, 1, 4, 8, and 12 h) after IV injection of 0.05 mmol/kg of Gd-Hex-LnP in nude mice. Data represent mean ± SD, number of mice per group = 3.

**Table 1 ijms-24-04288-t001:** Characterization in terms of hydrodynamic diameter, polydispersity index, zeta potentials and entrapment efficiency of LnP and Zhubech.

Formulation	Drug	Lipid Composition	Molar Ratio	Hydrodynamic Diameter (nm)	PDI	Zeta Potential (mV)	E.E (%)
LnP	-	DPPC: MPPC: Chol: DSPE-PEG_2000	50:25:20:5	90.0 ± 0.65	0.39 ± 0.01	−20.02 ± 8.5	-
Zhubech	MFU	DPPC: MPPC: Chol: DSPE-PEG_2000	50:25:20:5	124.9 ± 3.2	0.16 ± 0.005	−30.3 ± 12	97.2 ± 0.9

Data are expressed as mean ± SD.

**Table 2 ijms-24-04288-t002:** Physical stability in terms of temperature effects, changes in hydrodynamic diameter, polydispersity index, MFU content, and physical appearance of Zhubech.

Day	Temperature	Hydrodynamic Diameter (nm)	PDI	MFU Content in LnP (%)	Physical Appearance
0	25 ± 1.5 °C	124.9 ± 3.2	0.16 ± 0.005	97.2 ± 0.9	Bluish, clear
30 _(Batch 1)_	4 ± 1 °C	117 ± 2.0	0.18 ± 0.04	95.2 ± 2.7	Cream white, clear
	25 ± 2 °C	127.4 ± 2.3	0.23 ± 0.08	93.4 ± 3.7	Cream white, clear
	40 ± 3 °C	126.8 ± 0.9	0.47 ± 0.06	91.9 ± 0.2	Cream white, clear
60 _(Batch 2)_	4 ± 1 °C	129.4 ± 0.6	0.27 ± 0.013	90.1 ± 1.1	Cloudy
	25 ± 2 °C	135 ± 6.1	0.63 ± 0.031	87.1 ± 2.1	White, hazy
	40 ± 3 °C	142.7 ± 2.5	0.36 ± 0.021	90.3 ± 4.3	Cream white, clear
90 _(Batch 3)_	4 ± 2 °C	146.7 ± 1.6	0.39 ± 0.07	71.1 ± 0.8	Cloudy
	25 ± 2 °C	162.2 ± 16.0	0.42 ± 0.02	68.2 ± 3.7	White, hazy
	40 ± 3 °C	169.1 ± 2.1	0.15 ± 0.01	67.9 ± 6.4	Cream white, clear

Data are expressed as mean ± SD.

**Table 3 ijms-24-04288-t003:** Comparing the IC_50_ values of 5-FU compared to MFU and Zhubech in MiaPaca-2 and Panc-1 cell lines.

Structure	MiaPaca-2	Panc-1
2D	3D Spheroid	3D Organoid	*p*-Value	2D	3D Spheroid	*p*-Value
5-FU 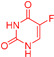	5.4 ± 1.1	8.5 ± 1.2	-	0.03 (5-FU_(2D)_ vs. MFU) *	3.0 ± 1.1	7.8 ± 1.1	0.31 (5-FU_(2D)_ vs. MFU)0.12 (MFU(2D) vs. Zhubech
MFU 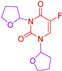	3.9 ± 1.0	6.7 ± 0.9	42.3 ± 1.0	0.01 (5-FU(3DS) vs. MFU) *	2.3 ± 0.8	6.8 ± 1.1	0.9 (5-FU_(3D)_ vs. MFU
Zhubech 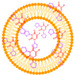	2.2 ± 1.1	5.0 ± 0.9	9.8 ± 1.4	<0.00001 (MFU(3DO) vs. Zhubech) *0.002 (5-FU _(2D)_ vs. Zhubech *	2.0 ± 0.7	3.4 ± 1.0	0.0003 (MFU_(3D)_ vs. Zhubech) *

Data are expressed as mean ± SD; * = *p*-value is significant; 3DO (3D organoid); 3DS (3D spheroid).

## Data Availability

Not applicable.

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
