# Peer review of "Evaluation of Anticancer Activity of Zhubech, a New 5-FU Analog Liposomal Formulation, against Pancreatic Cancer"

_ijms, 2023, doi:10.3390/ijms24054288_

Round 1

Reviewer 1 Report

This research work has been nicely carried out and presented by the authors, however, needs some major changes to be done before its consideration for publication.

Comments

1.     In introduction section (line 115-118) author mentioned in-vitro cell viability evaluation of developed formulation was carried out in both spheroids and organoids of both cell lines, but data of Panc-1 organoids is not presented in the manuscript. Reviewer suggests to present the cell viability data with Panc-1 organoids also for the comparative study with MiaPaca-2 organoids.

2.     In figure 1: the legend of figures and figures presented are confusing and not matching with the labelling on the vials. For instance, in figure 1 a) the vial label is zhubech powder but it is presented as MFU powder. Similarily in figure 1 f) the temperature condition on vial is written as 25°C while as per to legend it should be 40°C.

3.     Please reconsider the legend of table1, it should be in detail mentioning the data presented in table.

4.     Why there is so much difference in zeta potential value of LnP and Zhubech as there is no change in the surface charge or structure after MFU loading.

5.     The figure 3 b) the image presents 2D view of colocalization of LnP within the cytoplasm of the cells at different angle of rotation and is not time dependent while figure legend puzzling and not matching with the images presented. Please reconsider the legend of the figure.

6.     In line 229-330, the author mentioned the IC50 values of free MFU and zhubech treated 3D panc-1 cultures as 3.4 ± 1.0µM and 6.8 ± 1.1µM respectively. Please cross check the values presented in the section. Moreover, IC50 values written in the figures are not matching with the values written in text as well as in table 3. Please have cross check the values. 

7.     In figure 5, (cell viability graphs of different experiments), the x-axis shows the log concentration of MFU/5-FU. However, the consistency is not same for all graphs. Some of them represents concentration range from 0.0 – 2.0, while others range from 0.0 – 1.5. Please provide clarification.

8.     What do you mean by control group in section 2.3 tumor efficacy studies and if it is refers to MFU why it has low tumor inhibition capacity than 5-FU. And if it is not MFU then why MFU was not considered for this study.

9.     In figure 7, the biodistribution of Gd-Hex-LnP was much higher in liver and lungs as compared to pancreas and other tissues. It may cause accumulation of liposomal formulation at these tissues causing off target toxicity. The reviewer suggests to give proper justification for the same. And it would be better if the ligand is attached over the surface of the formulation to achieve target delivery to the tumor microenvironment reducing the adverse effects of MFU at non target organs. In addition, correct the figure x-axis which displays (time in hr), after 0.5 hr it should be 1 hr instead of 10 hr.

10. Please provide the TEM images of developed Zhubech formulation.

Author Response

Comments

  1. In introduction section (line 115-118) author mentioned in-vitro cell viability evaluation of developed formulation was carried out in both spheroids and organoids of both cell lines, but data of Panc-1 organoids is not presented in the manuscript. Reviewer suggests to present the cell viability data with Panc-1 organoids also for the comparative study with MiaPaca-2 organoids.

Response: We thank the reviewer and appreciate the generosity of their time in reviewing this study. Panc-1 cells were very fragile and could not form a good for organoid. As such we could only report 2D and 3D spheroids for Panc-1 cells. We have reworded lines 115-118 to suit the description of the study.

  1. In figure 1: the legend of figures and figures presented are confusing and not matching with the labelling on the vials. For instance, in figure 1 a) the vial label is zhubech powder but it is presented as MFU powder. Similarily in figure 1 f) the temperature condition on vial is written as 25°C while as per to legend it should be 40°C.

Response: We appreciate this observation. Changes have been made and Figure 1 updated.

  1. Please reconsider the legend of table1, it should be in detail mentioning the data presented in table.

Response: Table 1 legend updated (see table 1). 

  1. Why there is so much difference in zeta potential value of LnP and Zhubech as there is no change in the surface charge or structure after MFU loading.

Response: We obtained a mean zeta potential of -20.02±8.5 mV for LnP, an important omission was made and has been corrected. (See table table 1)

  1. The figure 3 b) the image presents 2D view of colocalization of LnP within the cytoplasm of the cells at different angle of rotation and is not time dependent while figure legend puzzling and not matching with the images presented. Please reconsider the legend of the figure.

Response: Thank you for the observation. This was a manipulation of the 3D view of the colocalization of LnP in Panc-1 cells (please see a 25 second video in supplemental video S1). In addition, we have reworded the Figure legend for clarity (See figure 3).

  1. In line 229-330, the author mentioned the IC50 values of free MFU and zhubech treated 3D panc-1 cultures as 3.4 ± 1.0µM and 6.8 ± 1.1µM respectively. Please cross check the values presented in the section. Moreover, IC50 values written in the figures are not matching with the values written in text as well as in table 3. Please have cross check the values. 

Response: Authors agree with the observation of the reviewer, changes have been made (See table 3)

  1. In figure 5, (cell viability graphs of different experiments), the x-axis shows the log concentration of MFU/5-FU. However, the consistency is not same for all graphs. Some of them represents concentration range from 0.0 – 2.0, while others range from 0.0 – 1.5. Please provide clarification.

      Response: Authors agree with the observation of the reviewer as Figure 5 seems to have a different concentration range. This is because we had to increase the concentration range of MFU for organoid studies (up to 100 µM) thereby extending the graph to 2.0 as in previous graph. (See Figure 5), meanwhile for other 2D and 3D spheroids concentration ranges ended at 40 µM. Reason being that from literature organoids usually would need higher dosage for treatments compared to 2D culture models[1, 2].

  1. What do you mean by control group in section 2.3 tumor efficacy studies and if it is refers to MFU why it has low tumor inhibition capacity than 5-FU. And if it is not MFU then why MFU was not considered for this study.

Response: Control here means untreated animals; this has been updated in Figure 6. In vitro studies (see Table 3) already demonstrated the effectiveness of MFU compared to 5-FU and based on this we proceeded to entrap MFU in the liposomal nanoparticles (zhubech) and compare its tumor suppression efficacy with the standard drug (5-FU) which is used in clinical practice.

  1. In figure 7, the biodistribution of Gd-Hex-LnP was much higher in liver and lungs as compared to pancreas and other tissues. It may cause accumulation of liposomal formulation at these tissues causing off target toxicity. The reviewer suggests to give proper justification for the same. And it would be better if the ligand is attached over the surface of the formulation to achieve target delivery to the tumor microenvironment reducing the adverse effects of MFU at nontarget organs. In addition, correct the figure x-axis which displays (time in hr), after 0.5 hr it should be 1 hr instead of 10 hr.

      Response: Authors agree with the reviewer. In our future studies, we will focus on developing and targeting zhubech using appropriate ligands to target pancreatic cancer to minimize toxicity and increase its uptake in pancreatic cancer cells. Corrections to the labeling of the X-axis of Figure 7 have been done.

  1. Please provide the TEM images of developed Zhubech formulation.

      Response: Thank you for the suggestion, authors do agree that it is important to include TEM. We will conduct TEM and SEM imaging in the development of targeted zhubech in our future studies.

Reference

[1]         M. Londoño-Berrio, C. Castro, A. Cañas, I. Ortiz, and M. Osorio, "Advances in Tumor Organoids for the Evaluation of Drugs: A Bibliographic Review," Pharmaceutics, vol. 14, no. 12, p. 2709, 2022. [Online]. Available: https://www.mdpi.com/1999-4923/14/12/2709.

[2]         S. Kim et al., "Comparison of Cell and Organoid-Level Analysis of Patient-Derived 3D Organoids to Evaluate Tumor Cell Growth Dynamics and Drug Response," (in eng), SLAS Discov, vol. 25, no. 7, pp. 744-754, Aug 2020, doi: 10.1177/2472555220915827.

Reviewer 2 Report

The current research is a good effort in targeting and enhanced retention of therapeutics to treat the Pancreatic cancer. Nkafu Bechem Ndemazie et al, put great efforts to conclude a diversified multifaceted project. However, room for improvement is always there and the study can further be refined by including following suggestions to make it appealing for a vast majority of readers.

Line NO                                       Comments

16- How and why the term Zhubech is used?

87- PeGylation offers systemic stealth effect due to surface hydration to facilitate enhanced delivery to the target site. Pl include

139- 1The type of container has greater effect of the stability profile of liposomes. So, mention the type of storage container, like plastic and its type, or glass and its type and information about light protection. 

2- The reference mentioned at 28 is recommended for oral suspensions, then  how it can be employed for liposomal stability. The following article is helpful regarding stability protocols of liposomes; 

Chloroform-Injection (CI) and Spontaneous-Phase-Transition (SPT) Are Novel Methods, Simplifying the Fabrication of Liposomes with Versatile Solution to Cholesterol Content and Size Distribution

https://doi.org/10.3390/pharmaceutics12111065 

140, Figure 1- If appearance is a parameter; then all photograph at each storage condition must be provided for comparison.

144- Write oC, as a standard symbol.

156, Table 1- The difference in composition must be mentioned, which consequently altered the physical properties.

159, Table 1- A- What you want to show from the results of Physical appearance.

B- If the liposomes were coagulated at 25oC, then how the remain at 40oC at day 60 and 90.

C- How the values of hydrodynamic diameter and PDI were measured with coagulated formulation.

D- Zeta potential is most important during stability studies as it demonstrates the intactness of drug loaded liposomes. Please include. 

171- Pl mention statistical measure for significance?

199- A- Typographic mistake

B- The caption of Figure 2, shows that the release of free MFU has been evaluated. While Zhubec is under question.

C- The text do not match the figure.

217- Typographic mistake

218- Typographic mistake

219- It would be better to analyze Rhodamine/MFP co-loaded particles in the cell culture and evaluate the result through flow cytometry to quantify percent internalization in a time dependent manner.

227- Reference is missing

271- incorporate full word before Abbreviation

295- discuss stealth effect 

298- "The liposomes are stable at room temperature, hence do not need special storage conditions" The stability study results do not support this stance.

318- typographic mistake

323- typographic mistake

354- write complete word before abbreviation

368- A- Mention the type of storage container, as it greatly effect the stability.

B- Zeta potential measurement is essential in drug loaded liposomes, as it illustrates intactness of the liposomes.

C- Zhubec have Zeta potential -30mv, how the hydrodynamic size be measured in DI water?

D- The results section shows accelerated stability only, why the real time stability is claimed in 4.3.. 

5 The protocols mentioned at Ref 28 and 47 are for Oral suspension and Transdermal patches respectively. How these protocols can be employed to a liposomal formulation?

6 It would be better to cryoprotect and freeze dry the liposomal formulation to enhance shelf life stability.

7  The following article is recommended to be cited in the discussion and introduction parts to address formulation related data;

Chloroform-Injection (CI) and

Spontaneous-Phase-Transition (SPT) Are Novel Methods, Simplifying the Fabrication of Liposomes with Versatile Solution to Cholesterol Content and Size Distribution

https://doi.org/10.3390/pharmaceutics12111065

283- Typographic mistake

399- Typographic mistake

414- A- Protocols used in the treatment of cell cultures may be included. 

B- Protocols used in the treatment of Animal models may be included. 

Author Response

Please see the attachment for the address of the reviewer's concerns.

Thank you

 Line NO                                       Comments

16- How and why the term Zhubech is used?

Response: Zhubech was coined from the names of 2 co-authors (Xue Zhu and Nkafu Bechem) who worked on the synthesis and formulation of MFU and LnP respectively. Zhubech is the term used for MFU-loaded liposomal formulation.

87- PeGylation offers systemic stealth effect due to surface hydration to facilitate enhanced delivery to the target site. Pl include

Response: The sentences have been revised to include the suggested statement. (see lines 95-96)

139- 1The type of container has greater effect of the stability profile of liposomes. So, mention the type of storage container, like plastic and its type, or glass and its type and information about light protection. 

Response: Authors did not use light sensitive materials for the preparation of zhubech. We stored samples in 20mL borosilicate glass vials material as shown in Figure 1 (see line 137)

2- The reference mentioned at 28 is recommended for oral suspensions, then how it can be employed for liposomal stability. The following article is helpful regarding stability protocols of liposomes; 

Response: We also have included more references for IV liposomal formulation and stability study which was in accordance with our methods (see Line 133).

Chloroform-Injection (CI) and Spontaneous-Phase-Transition (SPT) Are Novel Methods, Simplifying the Fabrication of Liposomes with Versatile Solution to Cholesterol Content and Size Distribution

https://doi.org/10.3390/pharmaceutics12111065 

Response: We agree with the reviewer. There are different methods for preparing liposomes, however, we focused on the film hydration method because our lab, over 10 years, has optimized this method for anticancer drugs through the IV route.  (see lines 85-87).

140, Figure 1- If appearance is a parameter; then all photograph at each storage condition must be provided for comparison.

Response: Authors have included a complete description of the physical appearances in Table 2, we included Figure 1 just to show what we had earlier mentioned in the paragraph. For example, all batch 1 samples (at different temperatures) appeared cream-white and clear, so image “d” of Fig 1 shows a representation. So, including all pictures will make Figure 1 very cumbersome. We summarized these findings in Table 2 and Figure 1.

144- Write oC, as a standard symbol.

Response: The statement has been corrected as proposed by the reviewer (see line 151)

156, Table 1- The difference in composition must be mentioned, which consequently altered the physical properties.

Response: The difference in LnP and zhubech is indicated in Table 1. Zhubech consists of MFU in the liposomal nanoparticles while LnP is just a blank liposomal nanoparticle. The studies focused only on the physical stability of zhubech at varying temperatures and measured the following parameters: hydrodynamic diameter, PDI, MFU content and physical appearance.

159, Table 1- A- What you want to show from the results of Physical appearance.

Response: Physical appearance is a secondary parameter to demonstrate the difference in the physical appearance of the formulations at different temperatures over time.

B- If the liposomes were coagulated at 25oC, then how the remain at 40oC at day 60 and 90.

Response: The term coagulated was inappropriately used in this context which has been changed to suspension and has been corrected in the manuscript (see Table 2, Line 141 and Figure 1). The appearance of the formulation at 400C for 60 days was cloudy while for 90 days it was white hazy (see table 2).

C- How the values of hydrodynamic diameter and PDI were measured with coagulated formulation.

Response: Again, the word coagulated was misapplied in this context, the appropriate term is hazy and the PDI values obtained were measured using the NICOMP particle size analyzer.

D- Zeta potential is most important during stability studies as it demonstrates the intactness of drug loaded liposomes. Please include. 

Response: We agree with the reviewer that zeta potential monitoring during the stability studies is one of the parameters to be considered. Per the literature and other published papers, we focused on the described parameters (hydrodynamic diameter, PDI, MFU content, and physical appearance) in our studies. We will include zeta potential as a parameter in the physical stability monitoring of our targeted zhubech in our future studies.

171- Pl mention statistical measure for significance?

Response: Statistical significance was measured as described in the statistical analysis section (see lines 542 to 546). The statement is revised to include statistical significance as suggested

199- A- Typographic mistake

Response: It has been corrected (See line 211)

B- The caption of Figure 2, shows that the release of free MFU has been evaluated. While Zhubech is under question.

Response: Very good observation, Figure 2 caption has been revised and Figure 2a has been updated

C- The text do not match the figure.

Response: Authors have included a description of Figure 2 in Lines 174-194, in addition, lines 179-180 have been reworded for clarity.

217- Typographic mistake

Response: It has been corrected (see line 230)

218- Typographic mistake

Response: Typographic error has been corrected. (see line 231)

219- It would be better to analyze Rhodamine/MFP co-loaded particles in the cell culture and evaluate the result through flow cytometry to quantify percent internalization in a time dependent manner.

Response:  We agree with the reviewer. We will compare the percent internalization of both zhubech and targeted zhubech using flow cytometry and confocal imaging in a time-dependent manner in our next studies.

227- Reference is missing

Response: Reference added please see reference number 34 on line 242.

271- incorporate full word before Abbreviation

Response: Inductively coupled plasma mass spectrometry (ICP-MS) was first mentioned and abbreviated in the abstract section (line 22). We have redefined it again as suggested by the reviewer (please see line 283).

295- discuss stealth effect 

Response: We agree and have included the statements on the stealth effect of PEGylated liposomes (please see lines 308-310)

298- "The liposomes are stable at room temperature, hence do not need special storage conditions" The stability study results do not support this stance.

Response: We agree and have reworded the sentence to give its full meaning (please see lines 313-317).

318- typographic mistake

Response: Authors could not identify any typographic error in the sentence (see line 344)

323- typographic mistake

Response: Authors could not identify any typographic error in the sentence (see line 349)

354- write complete word before abbreviation

Response: Authors have included the full meaning of PBS (Line 386)

368- A- Mention the type of storage container, as it greatly effect the stability.

Response: Authors have included the type of storage container (see lines 134 and 392)

B- Zeta potential measurement is essential in drug loaded liposomes, as it illustrates intactness of the liposomes.

Response: Zeta potential values of Lnp and zhubech have been included in Table 1:

C- Zhubec have Zeta potential -30mv, how the hydrodynamic size be measured in DI water?

Response: Hydrodynamic size refers to a measurement of the particle size of formulation in a solution or liquid. Also, the formulation can be diluted with DI and measured it’s hydrodynamic diameter.    

D- The results section shows accelerated stability only, why the real time stability is claimed in 4.3.

Response: This section has been updated for stability studies conducted based on temperature changes over time (see line 393). 

5 The protocols mentioned at Ref 28 and 47 are for Oral suspension and Transdermal patches respectively. How these protocols can be employed to a liposomal formulation?

Response: This has already been addressed and references 27 and 55 are cited to support the methods used in our study.

6 It would be better to cryoprotect and freeze dry the liposomal formulation to enhance shelf life stability.

Response: Freshly prepared formulations were used in our studies.  We plan to conduct freeze-dry or lyophilization using mannitol, sucrose, or lactose on zhubech and targeted zhubech to assess their stability (particle size, polydispersity, and zeta potential values) in our future studies. 

7 The following article is recommended to be cited in the discussion and introduction parts to address formulation related data;

Chloroform-Injection (CI) and

Spontaneous-Phase-Transition (SPT) Are Novel Methods, Simplifying the Fabrication of Liposomes with Versatile Solution to Cholesterol Content and Size Distribution

https://doi.org/10.3390/pharmaceutics12111065

Response: The above reference has been cited in the manuscript (See lines 85-87 and 306-309.

283- Typographic mistake

Response: we thoroughly reviewed this line and couldn’t see any typographical error

399- Typographic mistake

Response: We have thoroughly reviewed this line and could not find any typographical error

414- A- Protocols used in the treatment of cell cultures may be included.

Response: Authors agree with the reviewer, it was a great omission. The protocol for cell culture has been added (please see lines 449 to 494).

B- Protocols used in the treatment of Animal models may be included. 

Response: Authors agree with this suggestion from the reviewer. Protocol for animal studies has been added (see lines 495 to 541).

Reviewer 3 Report

The paper by Ndemazie et al. entitled “Evaluation of anticancer activity of zhubech, a new 5-FU analog liposomal formulation, against pancreatic cancer” presents the formulation and characterization of 1,3-bis tetrahydrofuran-2yl-5FU (MFU)-loaded liposomal nanoparticles (Zhubech). The authors further evaluate the stability, MFU release kinetics, and anticancer activities in patient-derived pancreatic cancer cells in vitro and tumor xenograft mouse models.

The authors conclude that they successfully formulated Zhubech, which showed reduced viability, two- or four-fold lower than FU-treated cells in 3D spheroids and organoids culture models. Tumor efficacy studies in PDX bearing mouse model showed more than a 9-fold decrease in mean tumor volumes when treated with Zhubech compared to 5-FU treated animals, demonstrating that Zhubech may be a potential candidate for delivering drugs for pancreatic cancer treatment. In general, the authors provided a substantial level of evidence in support of their findings in this manuscript. The manuscript presents remarkable experimental results, which may advance the understanding of innovative anticancer drug delivery. I have a few minor suggestions to improve the quality of the paper.

1.   Figures - Figure 5; authors should re-write fig. 5 legend for clarity. Repetition of ‘% viability…..’ in the legend is superfluous.

e.g., Figure 5: Cell viability: Effects of Zhubech formulation on Panc-1 and MiaPaca-2 cell growth after 48 h exposure of (a) Panc-1 in 2D culture, (b) Panc-1 in 3D spheroid culture, (c) MiaPaca-2 in 2D culture, and (d) MiaPaca-2 in 3D spheroid culture.

(e) Percentage viability of MiaPaca-2 in 3D organoid culture after 7 days of exposure with MFU and Zhubech formulation.

2.    Tables – Table 3; authors should check the fonts in table 3 for consistency – e.g., subscripts, abbreviations, etc.

Author Response

  1. Figures- Figure 5; authors should re-write fig. 5 legend for clarity. Repetition of ‘% viability…..’ in the legend is superfluous.

e.g., Figure 5: Cell viability: Effects of Zhubech formulation on Panc-1 and MiaPaca-2 cell growth after 48 h exposure of (a) Panc-1 in 2D culture, (b) Panc-1 in 3D spheroid culture, (c) MiaPaca-2 in 2D culture, and (d) MiaPaca-2 in 3D spheroid culture.

(e) Percentage viability of MiaPaca-2 in 3D organoid culture after 7 days of exposure with MFU and Zhubech formulation.

Response: Authors agree with the suggestions of the reviewer. Legend has been updated (Please see lines 249-251)

  1. Tables– Table 3; authors should check the fonts in table 3 for consistency – e.g., subscripts, abbreviations, etc.

Response: Authors agree, and Table 3 has been edited.

Reviewer 4 Report

Authors have presented a high value scientific research with novel hypotheses and outcomes. The objective of the research is well supported with various experiments and has significant impact. Overall, authors have synthesized a MFU and loaded into LNP formulation and presented in-vitro and in-vivo studies. They have concluded the superior tumor efficacy and cellular uptake of Zhubech over controls. Also, biodistribution data in liver and pancreas supported and justified all the efforts.  However, there are few general comments and suggestions to improve the final quality of the manuscript.

1.     The description about MFU content in the section 2.1.2 over the stability is not inline with the data in Table 2.

2.     Can you comment on why MFU content 4 C and 40 C are lower than 25 C, particularly for 2- and 3-months data?

3.     It looks like final Zhubech formulation is in the liquid form and as per Figure 1, over the time appearance changes which is poor stability sign and it may cause any significant changes like dose dumping/leakage, particle growth etc. In such case, why authors have not tried freeze drying approach?

4.     Figure 7, on x-axis, it should be 1.0 h and not as 10 h.

5.     For dialysis, authors have used mixture of PBS-Methanol as a media, why methanol? How can you justify presence of methanol with respect to the in vivo condition?

6.     In methods, animal study details for tumor efficacy and biodistribution are missing.

Author Response

Authors have presented a high value scientific research with novel hypotheses and outcomes. The objective of the research is well supported with various experiments and has significant impact. Overall, authors have synthesized a MFU and loaded into LNP formulation and presented in-vitro and in-vivo studies. They have concluded the superior tumor efficacy and cellular uptake of Zhubech over controls. Also, biodistribution data in liver and pancreas supported and justified all the efforts.  However, there are few general comments and suggestions to improve the final quality of the manuscript.

  1. The description about MFU content in the section 2.1.2 over the stability is not inline with the data in Table 2.

Response: Authors do appreciate the feedback from the reviewer, for the time put in to review and suggests changes to this manuscript. The EE was used to The term drug content was used interchangeably with entrapment efficiency (EE). (see lines 136-137)

  1. Can you comment on why MFU content 4 C and 40 C are lower than 25 C, particularly for 2- and 3-months data?

Response: Authors noted and described the change in MFU content and documents on Table 2. Here in, batch 3 corresponding to 90 days show a significant decrease in MFU content at all temperature.

  1. It looks like final Zhubech formulation is in the liquid form and as per Figure 1, over the time appearance changes which is poor stability sign and it may cause any significant changes like dose dumping/leakage, particle growth etc. In such case, why authors have not tried freeze drying approach?

Response: All viability and animal studies, the formulation was prepared fresh and used on the same day.

  1. Figure 7, on x-axis, it should be 1.0 h and not as 10 h.

Response: Agreed and corrected.

  1. For dialysis, authors have used mixture of PBS-Methanol as a media, why methanol? How can you justify presence of methanol with respect to the in vivo condition?

Response: We totally agree with the reviewer, we did use PBS for release kinetics. However, for entrapment efficiency the final product was mixed with H2O-Methanol (see line 432).

  1. In methods, animal study details for tumor efficacy and biodistribution are missing.

Response: Authors did omit these details when preparing the manuscript into the IJMS template format. This has been updated and we have included these sections (See Lines 438 to 531).

Reviewer 5 Report

The article entitled "Evaluation of anticancer activity of zhubech, a new 5-FU analog 2 liposomal formulation, against pancreatic cancer" demonstrates that Zhu-32 bech may be a potential candidate for delivering drugs for pancreatic cancer treatment and have done various studies to prove that, however, there are still some concerns.

1) In the background, can you please elaborate on the selection criteria of liposomes instead of another type of nanoparticles and pancreatic cancer instead another type of cancer?

2)Figure 5 data should be rearranged to make it more consistent with each other, the font and size of text and image should be considered.

3) The results should be discussed in more detail by referring to the previous studies, not only pointing out the findings but discussing those findings to make a sound conclusion.

4) Concluding remarks should be described in detail comprising conceptual insights from authors, future outlook, and limitations of the study.

Author Response

Response to Reviewer 5 Comments

The article entitled "Evaluation of anticancer activity of zhubech, a new 5-FU analog 2 liposomal formulation, against pancreatic cancer" demonstrates that Zhu-32 bech may be a potential candidate for delivering drugs for pancreatic cancer treatment and have done various studies to prove that, however, there are still some concerns.
1)    In the background, can you please elaborate on the selection criteria of liposomes instead of another type of nanoparticles and pancreatic cancer instead another type of cancer?
Response: Authors thank the reviewer for taking out their time to review and make suggestions for this research study. As concerns the selection of Liposomal formulation, authors have outlined in the introduction part of the research paper (lines 57 to 87). Well, we initially tested MFU in other cancer cell lines and was more effective in pancreatic cancer cell lines. The reason we are exploiting the effectiveness of MFU in Pancreatic cancer. Secondly 5-FU is one of the main drugs in the treatment of pancreatic cancer, so an analog which could address current drawbacks is imperative.
2)Figure 5 data should be rearranged to make it more consistent with each other, the font and size of text and image should be considered.
Response: Authors have formatted Figure 5 to increase the fonts in the figure.
3) The results should be discussed in more detail by referring to the previous studies, not only pointing out the findings but discussing those findings to make a sound conclusion.
Response: Authors agree and have included the link between the current studies and our previous studies (see lines 119-124, 237-238, and 270-271).
4) Concluding remarks should be described in detail, comprising conceptual insights from authors, future outlook, and study limitations.
Response: Authors agree and have included study limitations statements and future studies (See lines 540-542).

Round 2

Reviewer 1 Report

The authors have addressed the comments.